# Identification of SIRT3 modulating compounds in deep-sea fungi metabolites: Insights from molecular docking and MD simulations

**Abdullah R. Alanzi** [ID]*, **Bayan Abdullah Alhaidhal** [ID], **Raghad Mohammad Aloatibi**

Department of Pharmacognosy, College of Pharmacy, King Saud University, Riyadh, Saudi Arabia

* aralonazi@ksu.edu.sa

## Abstract

SIRT3, a crucial deacetylase that plays a key role in regulating mitochondrial acetylation, is tightly linked to metabolic processes and is essential for the maintenance of eukaryotic life. SIRT3 is a potential therapeutic target due to its key role in various diseases, including ageing, heart disease, cancer, and metabolic disorders. In this work, we aimed to identify potential SIRT3 inhibitors from the deep-sea fungal metabolites by employing molecular docking and ADMET analysis. Based on the binding affinities, ten compounds were selected whose docking scores were in the range of -9.693 to -8.327 kcal/mol. Further, four compounds Penipanoid C, Penicillactam, Quinolonimide, and Brevianamide R were selected based on the ADMET properties and subjected to Molecular dynamics simulations to assess the stability of these molecules with target. The stability analysis indicated that the selected compounds could act as lead compounds during *in vitro* assays to advance these drug candidates towards clinical drug development.

## 1. Introduction

Sirtuins are a family of protein deacetylases that depend on NAD + for their activity, and although they typically appear in only one or two forms in bacteria and archaea, they are present in seven distinct forms (SIRT1–7) in mammals [1–3]. These enzymes may be divided into five sub classes according to the conserved catalytic core domain. SIRT1, SIRT2, and SIRT3 are class I proteins that are identified by their high deacetylase activity levels. SIRT4 is a Class II enzyme that functions as an ADP-ribosyl transferase in mitochondria. While, SIRT5 is categorized as Class III, and SIRT6 and SIRT7 as Class IV [4,5].

SIRT3 is mostly found in the mitochondrial matrix and is essential for controlling various aspects of mitochondrial metabolism [6]. Its actions include the regulation of the following processes: the electron transport chain (ETC), oxidative phosphorylation (OXPHOS), fatty acid oxidation, amino acid metabolism, mitochondrial dynamics,

**Data availability statement:** All data has been incorporated with in the manuscript.

**Funding:** Authors extend their appreciation to researchers supporting project Number (RSPD2025R885) at King Saud University Riyadh Saudi Arabia for supporting this research.

**Competing interests:** The authors have declared that no competing interests exist.

detoxification of reactive oxygen species (ROS), urea cycle, tricarboxylic acid (TCA) cycle, and mitochondrial unfolded protein response (UPR) [7,8]. SIRT3 protein has been found in the brain, kidney, liver, heart, and other tissues with high mitochondrial concentrations [9,10]. The maintenance of mitochondrial function in these tissues is dependent on acetylation modifications mediated by SIRT3. The importance of mitochondria may be attributed to their vital functions in energy production, metabolism, intracellular signaling, and apoptosis. This is particularly true in tissues with high metabolic activity, which are more vulnerable to mitochondrial dysfunction [11]. Studies have shown the involvement of SIRT3 in the regulation of ageing, neurological disorders, cardiovascular disorders, hepatic disorders, kidney disorders, and further metabolic irregularities. Moreover, studies have shown its intriguing dual involvement in the development of cancer [10]. It also regulates the mitochondrial protein acetylation during renal fibrosis [12]. As a result, SIRT3 is a promising target for therapeutic intervention in a variety of human diseases [13]. Preclinical research on different small-molecule drugs that target SIRT3 has produced encouraging findings [14,15].

Currently, there is no widely recognized strong inhibitor or activator of SIRT3 suitable for therapeutic treatments. Still, a number of small-molecule compounds have been found that may have therapeutic value [2]. In particular, the well-known SIRT3 activator honokiol shows promise for treating illnesses linked to inflammation, cardiac problems, cancer, and metabolic issues. It's crucial to remember, that SIRT3 activation is not the only factor contributing to Honokiol's positive benefits [15]. Recent work by Upadhyay et al. indicates that honokiol stimulates SIRT3 through a non-allosteric mechanism. By extending the earlier findings of Guan et al., this work advances our knowledge of the molecular mechanisms behind Honokiol's SIRT3 activation [16]. Currently, 3-TYP and other SIRT3 inhibitors are mostly used as research purpose and are not yet utilized in therapeutic contexts [2,17]. This emphasizes how difficult it is yet to create strong SIRT3 activators or inhibitors for medical applications. Using scientific drug design techniques becomes imperative. Therefore, the purpose of this study was to screen deep-sea fungal metabolites to find new inhibitors that target SIRT3. The goal of the study is to find molecules that may open the door for the creation of therapeutic medicines that target pathways connected to SIRT3.

## 2. Methodology

### 2.1. Preparation of ligands

The structures of deep-sea fungal metabolites were downloaded from PubChem (https://pubchem.ncbi.nlm.nih.gov/). The Maestro LigPrep tool was utilized to prepare the structures of the phytochemicals [18]. Thirty-two conformers were produced for each ligand after the geometry of each ligand was optimized to assure accuracy. To reduce the compounds' energy, the OPLS_2005 force field was used [19].

### 2.2. Molecular docking

This study focuses on investigating the molecular interactions between deep-sea fungi metabolites and the SIRT3 protein to identify potential compounds with

inhibitory activity. The Protein Data Bank provided the crystal structure of SIRT3 (PDB ID: 4JSR). The SIRT3 protein structure was processed for docking using the Protein Preparation Wizard in Schrodinger Maestro [20]. Different steps were conducted, including the addition of hydrogens, the formation of disulfide bonds, the formation of zero-order bonds to metals, and the assignment of bond orders. PROPKA was used to eliminate extra ligands and water molecules and optimize hydrogen bonds at pH 7.0 [21,22]. Finally, the protein's energy was minimized using the OPLS_2005 force field [19]. A 3D grid was created at the binding site coordinates (X = 23.18, Y = 45.84, Z = -2.01) after the protein was prepared to aid in site-specific docking. Next, using the Glide docking module in SP (Standard Precision) mode, the prepared ligands were docked against SIRT3 [23]. Following analysis of the docked ligands, the Glide scores were used to determine the selection. Moreover, docking results were validated by re-docking procedure.

### 2.3. Toxicity analysis

The drug erosion is linked to toxicity concerns and suboptimal pharmacokinetics of the compounds [24]. To address this, the ADMET profiles were analyzed to evaluate the toxicity risks of drug candidates [25]. This predictive approach also helps in evaluating the likelihood of lead compounds becoming viable oral drugs. In this study, we used OSIRIS Property Explorer tool [26] to predict the ADMET characteristics of the most promising compounds. We assessed several pharmacokinetic properties, including molecular weight (MW), solubility (log S), logP, TPSA, and drug-likeness and score. Additionally, we scrutinized the compounds for potential toxicity consequences, encompassing tumorigenic, mutagenic, irritating, and reproductive concerns.

### 2.4. Molecular dynamics simulation and post MD analyses

The complexes were analyzed for protein conformation and ligand stability by running a simulation of 100 ns by using Desmond [27]. The systems were solvated by placing them in an orthorhombic box of 10 Å, filled with TIP3P water model [28]. To replicate physiological conditions, counter ions were added for neutralization, and 0.15 M NaCl salt was incorporated. The temperature and pressure of the system was set to 300K and 1 ATM respectively by using NPT ensemble. After a relaxation phase, the production run was started to record the trajectories after 50 ps time interval. The simulation interaction diagram module of Desmond was used to analyze the trajectories. In MD simulations with Desmond, we employed the standard Desmond relaxation protocol prior to the production run. This protocol involved an initial energy minimization followed by a short equilibration phase of 1 ns during which position restraints (using a force constant of 25 kcal/mol/Å²) were applied to the heavy atoms of the protein. This equilibration step allowed the solvent and ions to relax around the restrained protein-ligand complex. MMGBSA analysis was performed to calculate binding free energy calculations as well as hydrogen bonding and radius of gyration analyses were performed. PCA and 2D Free Energy Surface analyses was carried out to assess conformational variance and stability. Per-residue energy decomposition was carried out to estimate the individual binding free energy contribution of SIRT3 binding site residues to the stabilization and affinity of studied compounds.

## 3. Results

### 3.1. Molecular docking

The compounds identified through virtual screening were prepared, and a docking study was subsequently performed on the SIRT3 protein [29]. The binding affinities of all docked compounds were analyzed and then top ten compounds with binding affinities ranging from -9.693 to -8.327 kcal/mol were selected for further analysis (Table 1). The docking scores of the selected compounds suggested that these have potential for inhibiting the function of the SIRT3 protein. These findings provide valuable insights into the possibility of these compounds interacting effectively with the SIRT3 binding site, thereby warranting additional investigation into their inhibitory potential. Moreover, to further validate the docking analysis,

**Table 1. The glide scores of the docked compounds against SIRT3 protein.**

| Sr. | Compounds | PubChem CIDs | Glide score (kcal/mol) |
| --- | --- | --- | --- |
| 1 | Cyclopiamide D | 122218854 | −9.693 |
| 2 | Penilumamide K | 156580691 | −9.5 |
| 3 | Coccoquinone A | 132512004 | −9.09 |
| 4 | Penipanoid C | 136171438 | −8.86 |
| 5 | Penicillactam | 57337607 | −8.827 |
| 6 | Quinolonimide | 11492325 | −8.798 |
| 7 | Prenylterphenyllin | 23630784 | −8.751 |
| 8 | Brevianamide R | 49831335 | −8.553 |
| 9 | Butanolide A | 156581743 | −8.494 |
| 10 | Dichotocejpin C | 139583288 | −8.327 |

a re-docking procedure was performed. Interestingly, the both the native and redocked ligands' conformation and pose was somewhat highly similar in the binding site of the SIRT3 receptor validating the initial docking procedure [Fig 1].

### 3.2. ADMET analysis

The ADMET and toxicity risks profiles of the selected compounds were analyzed by OSIRIS Property Explorer tool, and it was observed that the predicted values were in the acceptable range (Table 2). The molecular weight plays a vital role in the distribution of a compound within cells, with lower-weight compounds generally able to distribute more easily throughout the body compared to those with higher weights [30]. To address this, a threshold of 500 g/mol was established, and all selected compounds fell within this range. cLogP determines the hydrophilicity of compound, a value of cLogP > 5 indicates poor absorption. The selected hits had cLogP values less than 5, indicating good absorption of compounds. The TPSA relates with the hydrogen bonding of a compound and is a good predictor of bioavailability [31]. TPSA < 160 Å$^2$ shows that the compound will have good oral bioavailability [32]. The hits had TPSA values in the range of 66.58 to 190.9 Å$^2$. Solubility is also a crucial factor in pharmacokinetics, influencing both the absorption and distribution of a compound. It is typically quantified as the logarithm of the solubility, expressed in mol/dm$^3$. This measurement helps to assess how easily a compound dissolves in a solvent, which is vital for its effective utilization in the body and its overall pharmacokinetic profile. The drug score is a comprehensive measure that combines several factors such as, cLogP, logS, molecular weight, and toxicity risk into a single, easy-to-understand number. This score is used to evaluate a compound's overall potential to become a drug. A higher drug score indicates a greater likelihood that the compound could be a viable drug candidate. In essence, the higher the drug score value, the more likely the compound is to be considered for further drug development [33]. Furthermore, the toxicity profile of compounds was evaluated, and it was observed that the compounds did not show toxicity tendencies except for **Cyclopiamide D, Coccoquinone A, Butanolide A,** and **Dichotocejpin C.** Based on the pharmacokinetic and toxicity profiles, 4 compounds, i.e., **Penipanoid C, Penicillactam, Quinolonimide**, and **Brevianamide R** were selected for molecular interactions analysis.

### 3.3. Molecular interactions analysis

The docked poses of the selected compounds were analyzed by Discovery Studio client tool to find the molecular interactions. The molecular interactions mainly involved hydrogen bonding, van der Waal interactions, pi-pi stacking, pi-sigma interactions, and alkyl (hydrophobic) interactions. The molecular interactions of each candidate compounds help in determining the binding affinities. Especially, the hydrogen bonds among ligand and protein atoms play an important role in the strength of protein-ligand complex [34]. **Penipanoid C** formed one van der Waal interaction with Ile291, two hydrogen

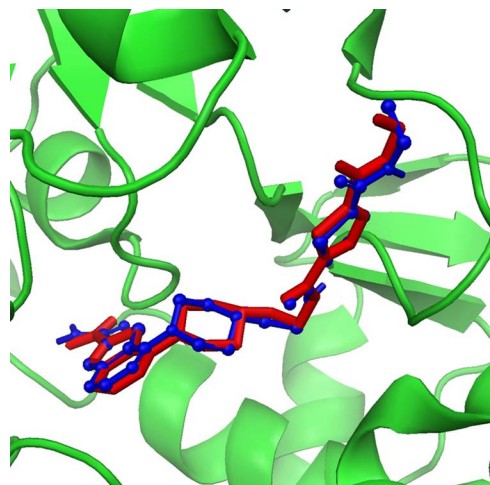

**Fig 1. Redocking of the ligand, native ligand is represented by red and redocked ligand is represented by blue colors.**

Table 2. The ADMET and Toxicity risks analysis of top ten compounds.

| Compounds | Pharmacokinetic Properties | | | | | | Toxicity Risks | | | |
|---|---|---|---|---|---|---|---|---|---|---|
| | MW | cLogP | TPSA | LogS | Drug likeness | Drug score | Mutagenic | Tumorigenic | Irritant | Reproductive effect |
| Cyclopiamide D | 336 | 2.28 | 83.55 | −4.49 | −3.52 | 0.08 | High | High | High | Passed |
| Penilumamide K | 442 | −1.16 | 190.9 | −3 | 2.72 | 0.76 | Passed | Passed | Passed | Passed |
| Coccoquinone A | 428 | 2.71 | 158.4 | −4.78 | −2.09 | 0.13 | High | Passed | High | Passed |
| **Penipanoid C** | **266** | **1.0** | **78.76** | **−3.06** | **4.35** | **0.9** | **Passed** | **Passed** | **Passed** | **Passed** |
| **Penicillactam** | **395** | **0.02** | **133.8** | **−3.59** | **6.56** | **0.8** | **Passed** | **Passed** | **Passed** | **Passed** |
| **Quinolonimide** | **228** | **−0.06** | **66.58** | **66.48** | **6.25** | **0.95** | **Passed** | **Passed** | **Passed** | **Passed** |
| Prenylterphenyllin | 406 | 5.82 | 79.15 | −5.87 | −0.23 | 0.27 | Passed | Passed | Passed | Passed |
| **Brevianamide R** | **379** | **3.03** | **74.43** | **−4.17** | **1.96** | **0.68** | **Passed** | **Passed** | **Passed** | **Passed** |
| Butanolide A | 254 | 1.85 | 66.76 | −2.53 | −10.55 | 0.28 | Passed | Passed | High | Passed |
| Dichotocejpin C | 262 | −0.63 | 81.08 | −1.17 | 6.65 | 0.35 | Passed | High | High | Passed |

bonds with Val292, Phe157, two Pi-Pi stacked interactions with His248, Phe180 and two alkyl interactions with Ile230, Ala146 (**Fig 2a**). **Penicillactam** formed five van der Waal interactions with Ile154, Pro155, Arg158, Gln228, Ile230, three hydrogen bonds with Phe157, Asn229, Tyr165, Pi-Sigma interaction with Ala146, and one alkyl interaction with His248 (**Fig 2b**). Similarly, **Quinolonimide** made two van der Waal interactions with Phe294, Ile154, two hydrogen bonds with Ile230, Phe157, two carbon hydrogen bonds with Gln228, Asp156, and two alkyl interactions with His248, Ala146 (**Fig 2c**). Lastly, **Brevianamide R** made one Pi-Sigma interaction with His248, two hydrogen bonds with Val292, Tyr165, one Pi-Pi stacked interaction with Phe294 and six alkyl interactions with Ile230, Ile291, Phe180, Phe157, Arg158, Ala146 (**Fig 2d**). The interactions of the remaining compounds are displayed in Table 3.

### 3.4. Analysis of plausible binding modes

The plausible binding modes of the selected compounds were observed by alignment on co-crystal ligand. This alignment showed that the docked poses of the hits occupied the same space in binding sites of the SIRT3 as occupied by co-crystal ligand (**Fig 3**). As a result, the plausible binding modes of the docked hits were evaluated for stability through Molecular

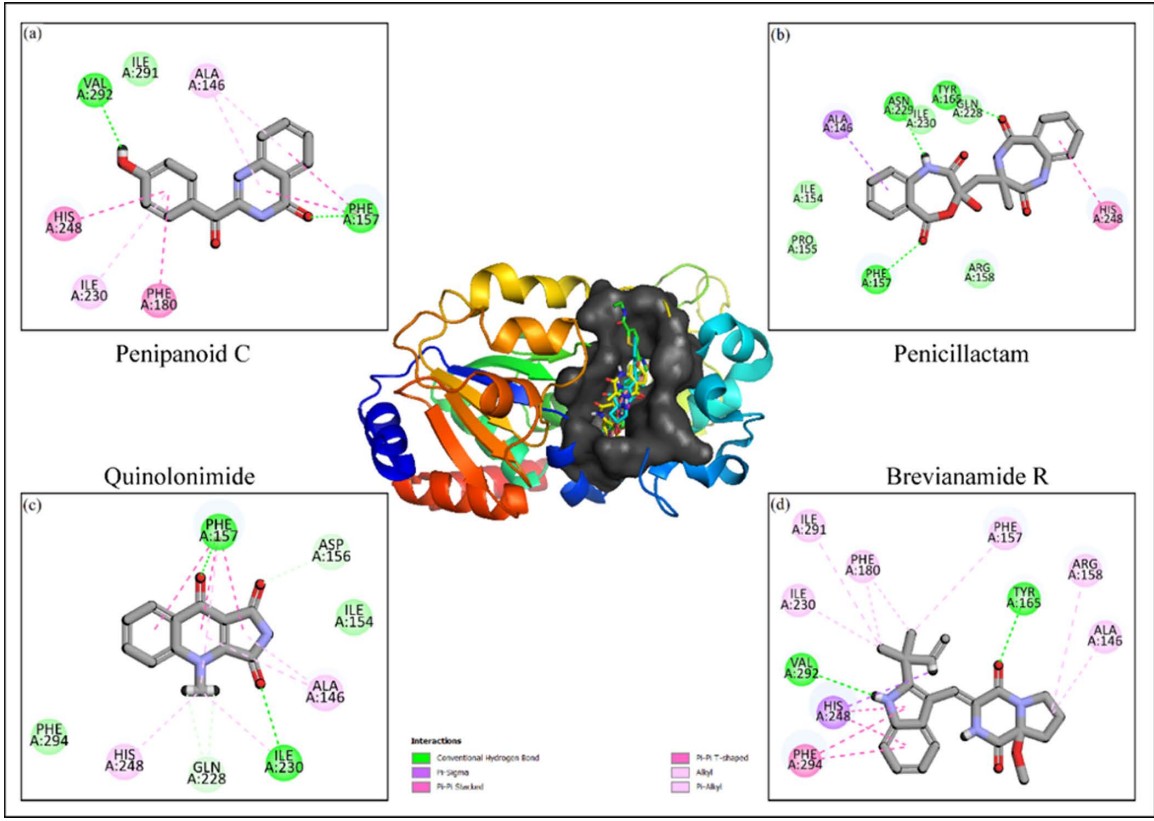

**Fig 2. The 2D molecular interactions of the selected compounds based on the pharmacokinetic and toxicity profiles with the SIRT3 receptor with diversity of bond formations indication a stabilization of ligands into receptor's active pocket.**

Dynamics (MD) simulations. In addition, key interactions such as hydrogen bonds with critical residues, e.g., Phe157, Val292, TYR165 and His248 and hydrophobic contacts were observed in the docked poses, mirroring those found in the co-crystallized structure. These observations support the plausibility of the binding modes predicted by docking, suggesting that the hit compounds engage the receptor in a manner consistent with known SIRT3 inhibition mechanisms. The stability of these binding orientations was subsequently assessed through molecular dynamics simulations, which further confirmed that the compounds maintained their key interactions and overall binding conformations throughout the simulation period.

### 3.5. MD simulation

**3.5.1. RMSD.** The root mean square deviation of the Cα atoms was measured to evaluate the structural changes of the complexes the RMSD values were calculated across the simulation [35,36]. The **Penipanoid C** complex's RMSD values displayed significant deviations during the first 20 ns, reaching a maximum value of 3.6 Å. However, after 20 ns, stability was reached in the 2.4 Å range, which was maintained throughout the simulation. The RMSD of ligand atoms was also aligned on protein atoms after 20 ns (**Fig 4a**). The RMSD of **Penicillactam** maintained the RMSD value of 2.1 Å at the start and remained in this range throughout the simulation., while the RMSD of ligand atoms aligned on protein atoms after 20 ns (**Fig 4b**). In the case of **Quinolonimide** complex, the RMSD of protein carbon-alpha atoms gradually increased to 2.8 Å at 10 ns and then maintained this range till 60 ns. The simulation ended with the RMSD staying in the

**Table 3. The molecular interactions of the selected compounds with SIRT3 protein.**

| Sr. | Compounds | Interactions |
|---|---|---|
| 1 | Cyclopiamide D | **Van der Waal:** Ala146, Tyr165, Gln228, Ile154, Ser149<br>**Hydrogen Bond:** Phe157, Ile230, Val292<br>**Alkyl:** His248, Phe180 |
| 2 | Penilumamide K | **Van der Waal:** Tyr165, Asn229<br>**Hydrogen Bond:** Val292, Ile230, Phe157<br>**Pi-Sigma:** Ala146<br>**Alkyl:** His248, Ile291 |
| 3 | Coccoquinone A | **Van der Waal:** His248, Val292, Phe294, Phe180<br>**Hydrogen Bond:** Gln228, Ile230, Ile154, Phe157, Tyr165<br>**Alkyl:** Ala146 |
| 4 | Penipanoid C | **Van der Waal:** Ile291<br>**Hydrogen Bond:** Val292, Phe157<br>**Pi-Pi Stacked:** His248, Phe180<br>**Alkyl:** Ile230, Ala146 |
| 5 | Penicillactam | **Van der Waal:** Ile154, Pro155, Arg158, Gln228, Ile230<br>**Hydrogen Bond:** Phe157, Asn229, Tyr165<br>**Pi-Sigma:** Ala146<br>**Alkyl:** His248 |
| 6 | Quinolonimide | **Van der Waal:** Phe294, Ile154<br>**Hydrogen Bond:** Ile230, Phe157<br>**Carbon Hydrogen Bond:** Gln228, Asp156<br>**Alkyl:** His248, Ala146 |
| 7 | Prenylterphenyllin | **Hydrogen Bond:** Phe293, Glu296, Gly295, Phe157<br>**Carbon Hydrogen Bond:** Phe180, Val292<br>**Alkyl:** Phe294, His248, Val324, Ile154, Leu199, Ile230, Ala146 |
| 8 | Brevianamide R | **Hydrogen Bond:** Val292, Tyr165<br>**Pi-Sigma:** His248<br>**Pi-Pi stacked:** Phe294<br>**Alkyl:** Ile230, Ile291, Phe180, Phe157, Arg158, Ala146 |
| 9 | Butanolide A | **Van der Waal:** Ile291, Tyr165, Arg158<br>**Hydrogen Bond:** Val292, Phe157, Phe231, Ile230<br>**Alkyl:** Phe294, Phe180, His248, Ala146 |
| 10 | Dichotocejpin C | **Van der Waal:** His248<br>**Hydrogen Bond:** Phe157, Ile154, Pro155, Ile230, Asn229, Gln228<br>**Pi-Cation:** Arg158 |

range of 3.2 Å after 60 ns. The RMSD of ligand atoms was less than protein atoms after 60 ns (**Fig 4c**). Finally, the RMSD of the **Brevianamide R** complex reached stability in the range of 2.4 at 20 ns and remained there until the end of the simulation. The RMSD of ligand atoms was aligned with RMSD of protein atoms (**Fig 4d**).

**3.5.2. RMSF.** Root mean square fluctuations (RMSF) values were computed in order to study the dynamic behavior of the proteins while they were bound to the ligands [37,38]. Throughout the simulation, the RMSF values provide information about the mobility and flexibility of specific protein residues. Most protein residues showed only slight changes during the simulation, measuring less than 1.8 Å, based on the estimated RMSF values. This suggests that while the ligands were present, these residues were stable and stiff. In contrast, higher RMSF values (4.8 Å) were found in the protein's loop regions (**Fig 5**). The bulk of residues in the protein-ligand complex maintained a rigid conformation, which indicated overall stability based on the Root Mean Square Fluctuation (RMSF) analysis. The loop regions exhibited higher RMSF values, indicating more prominent fluctuations and dynamic interactions with the ligands.

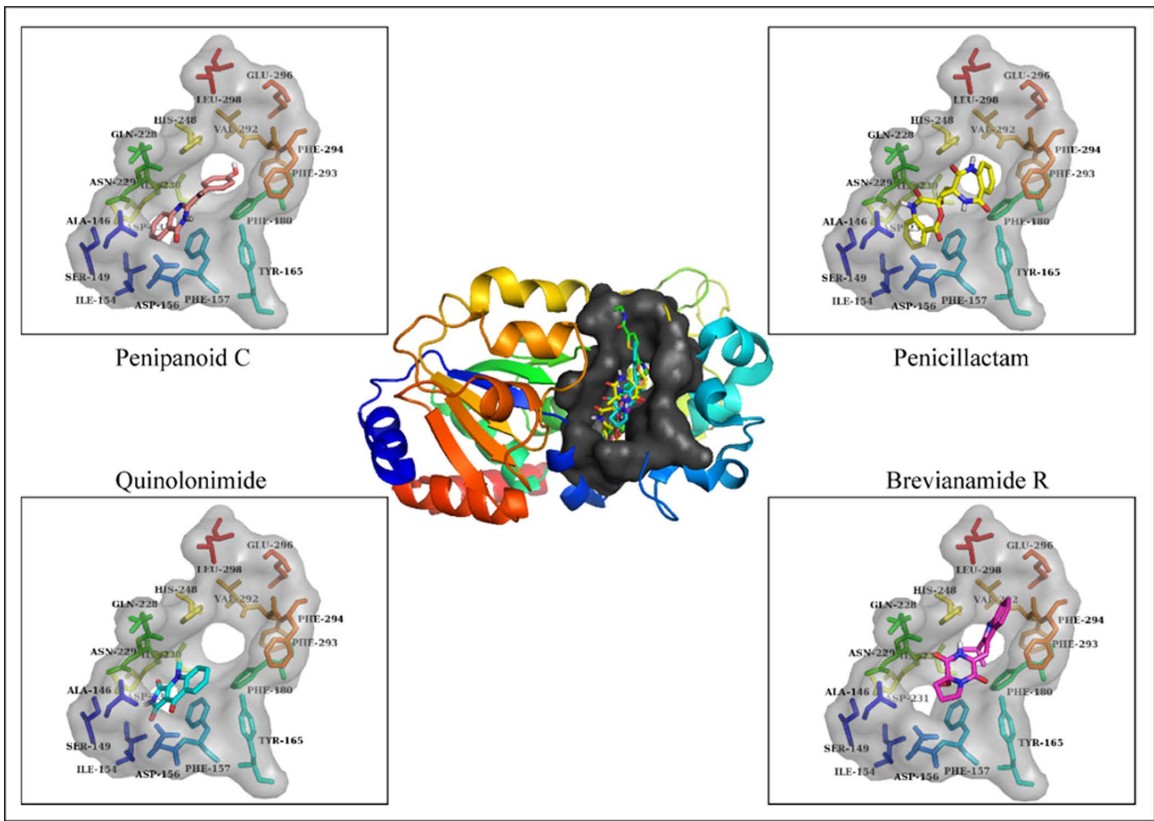

**Fig 3. The plausible binding poses of the top four compounds.** The binding modes of the compounds with the residues are also shown in sticks.

**3.5.3. Protein-ligand interactions.** During the simulation, the protein–ligand interfaces were predominantly stabilized by hydrogen bonds, water-mediated bridges, electrostatic forces, and hydrophobic contacts. These contacts play an important role in protein-ligand complex stability during the simulation. There residues involved in hydrogen bonding with **Penipanoid C** were Ser149, Ile154, Phe157, Gln228, Asn229, Ile230, and His248 (**Fig 6a**). In the **Penicillactam** complex, the residues involved in hydrogen bonding were Phe157, Tyr165, Gln228, Ile230, Asp231, and His248 (**Fig 6b**). In the **Quinolonimide** complex, the hydrogen bonding interactions involved Ile230 and Asp231 (**Fig 6c**). Lastly, the hydrogen bonding in **Brevianamide R** was observed in the following residues: Phe157, Leu164, Gln228, and His248 (**Fig 6d**). The hydrogen bonding during MD simulations provides important insights into the specific residues that stabilize the protein-ligand complexes. This information enhances our understanding on how these interactions contribute in the complex stability and binding affinities [39].

**3.5.4. Hydrogen bonding analysis.** Across the 100 ns simulation, each of the four complexes exhibited characteristic hydrogen-bonding patterns, reflecting both the ligand's affinity for SIRT3 and its dynamic fluctuations. Penipanoid C maintained a fairly consistent range of 13 hydrogen bonds over time, suggesting stable polar interactions within the active site. Penicillactam showed more pronounced fluctuations (spanning 0–4 hydrogen bonds), indicating transient formation and disruption of specific hydrogen-bond contacts. Quinolonimide likewise oscillated between 1–4 hydrogen bonds but displayed occasional spikes of higher bonding, hinting at short-lived yet strong polar contacts. Brevianamide R typically formed 1–3 hydrogen bonds, suggesting it largely retains a stable binding orientation while occasionally sampling alternative configurations (**Fig 7**). Collectively, these results confirm that all four ligands can sustain meaningful

**Fig 4. The RMSD plots of the SIRT3 complexes.** (a) Penipanoid C with initial deviations at about 20ns but then a good stabilization till the end, (b) Penicillactam with initial deviations at about 20ns but then a good stabilization till the end alongside the protein atoms, (c) Quinolonimide showed gradual increase in RMSD, (d) Brevianamide R.

hydrogen-bond interactions throughout the simulation, thus reinforcing the notion of stable binding modes inferred from the docking and other MD analyses.

**3.5.5. Radius of gyration.** For all four SIRT3-ligand complexes, the radius of gyration (Rg) analysis indicates that the overall protein compactness remains remarkably stable over the simulation. In detail, the Rg values for the SIRT3 complexes with Penipanoid C, Penicillactam, Quinolonimide, and Brevianamide R fluctuated within narrow margins (with deviations typically under 0.5 Å), suggesting that none of the ligands induce significant global structural changes in SIRT3. Notably, minor differences in the average Rg values were observed among the complexes, with the Penipanoid C and Brevianamide R complexes showing a slightly more compact protein conformation compared to the Penicillactam and Quinolonimide systems (**Fig 8**). These results confirm that the binding of these deep-sea fungal metabolites does not compromise the structural integrity of SIRT3 during the MD simulation, supporting their potential as stable inhibitors.

**3.5.6. PCA and 2D PCA-based free energy surface analysis.** For Penipanoid C, the PCA was performed to capture the primary conformational motions of the protein–ligand complex, with the first three eigenvectors explaining roughly 30%

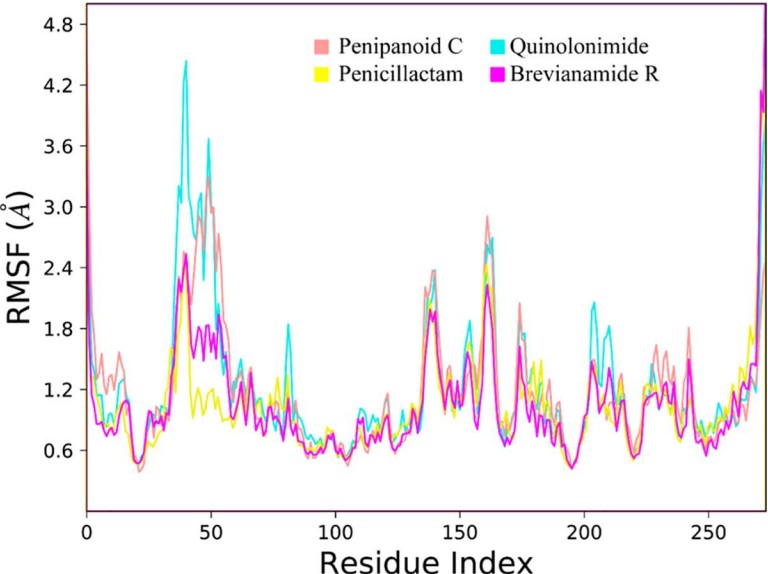

**Fig 5. The comparative RMSF plots of the SIRT3 protein in the presence of co-crystal ligand and hit compounds.**

of the total variance. PC1 showed the largest fluctuations, indicating the main direction of motion (**Fig 9A**). Additionally, the 2D PCA-based free energy surface revealed two main conformational clusters: a stable low-energy basin, where Penipanoid C predominantly resides, and a higher-energy region sampled less frequently. This distribution reflects the ligand's intrinsic flexibility while maintaining a largely favorable binding orientation throughout the simulation (**Fig 9B**). The PCA for the Penicillactam-SIRT3 complex revealed that the first few eigenvectors captured the primary motions of the system, with PC1 encompassing the largest fluctuations (**Fig 10A**). In the 2D PCA-based free energy surface, two main conformational clusters emerged: one low-energy basin where Penicillactam remained most of the time, and another higher-energy region visited less frequently (**Fig 10B**). This indicates that Penicillactam predominantly adopts a stable binding pose while occasionally sampling alternative configurations under physiological conditions. The principal component analysis (PCA) was performed to evaluate the conformational variance in the Quinolonimide-SIRT3 system. The first few eigenvectors contributed the most significant portion of the total variance, with PC1 capturing the highest degree of fluctuations (**Fig 11A**). A 2D PCA-based free energy surface was then generated by projecting the simulation data onto PC1 and PC2, revealing two main basins: a predominant low-energy region, where Quinolonimide remained most frequently, and a smaller high-energy state sampled intermittently (**Fig 11B**). The majority of frames were located in the local-minima well, indicating that Quinolonimide largely maintains a stable binding orientation while occasionally exploring alternative conformations under the simulation conditions. In the principal component analyses for Brevianamide R complex, the first three principal components (PC1, PC2, and PC3) accounted for 18.59%, 13.34%, and 9.72% of the total variance, respectively, giving a combined 41.65% coverage of overall conformational fluctuations (**Fig 12A**). PC1 thus showed the most significant motion at 18.59%. Furthermore, a 2D PCA-based free energy surface was generated to pinpoint the thermodynamically stable states, revealing a dominant low-energy basin where the complex spent most of its time, along with a higher-energy region sampled intermittently. This distribution suggests that the system predominantly adopts a stable conformation, with occasional excursions into alternative states (**Fig 12B**). These findings indicate that the system remains largely in a stable conformation while exploring higher-energy states less frequently, suggesting a well-defined, low-energy binding mode with limited but noteworthy conformational flexibility

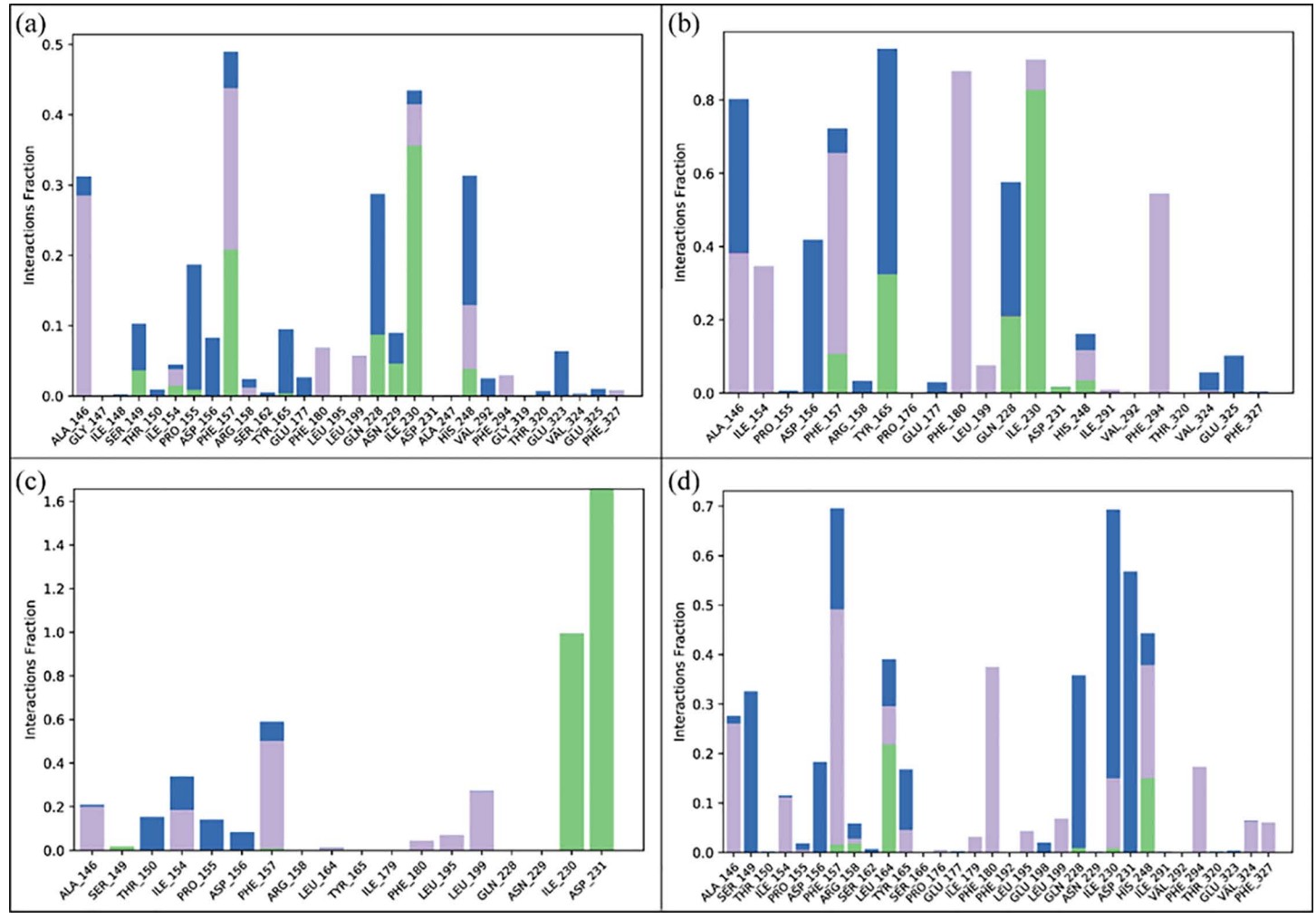

**Fig 6. The protein-ligand interactions.** (a) Penipanoid C, (b) Penicillactam, (c) Quinolonimide, (d) Brevianamide R. Green bars show hydrogen bonding, gray show hydrophobic interactions, and blue shows the water bridges.

**3.5.7. MMGBSA and energy decomposition analyses.** The MMGBSA results indicate that the van der Waals ($\Delta E_{vdW}$) and electrostatic ($\Delta E_{ele}$) terms are the primary contributors to binding across all four complexes, whereas the polar solvation energy partially offsets these favorable interactions. Notably, Quinolonimide and Penipanoid C exhibit slightly more negative total free energies ($\Delta G_{total}$), suggesting stronger overall affinity for SIRT3 compared to Penicillactam C and Brevianamide R. In each case, the negative gas-phase energy ($\Delta G_{gas}$) underscores stable, energetically favorable interactions in the binding pocket, while the solvation term ($\Delta G_{solv}$) exerts a smaller, unfavorable effect. These findings support the stable binding modes observed in molecular docking and MD simulations, further highlighting the potential of these compounds as SIRT3 inhibitors (**Fig 13**). Per residue energy decomposition analysis reveals that Phe157 consistently provides strong stabilizing interactions across all four complexes. For Penipanoid C, additional key contributions stem from Phe180, Ile230, and His248, reflecting strong hydrophobic and/or π-stacking contacts. Penicillactam likewise shows major energy contributions from Phe157, Phe180, and His248, with a moderate role from Ala146. In Quinolonimide, the largest negative energy values localize around Phe157 and Ile230, highlighting these residues as principal anchoring points. Brevianamide R, on the other hand, exhibits pronounced stabilization from

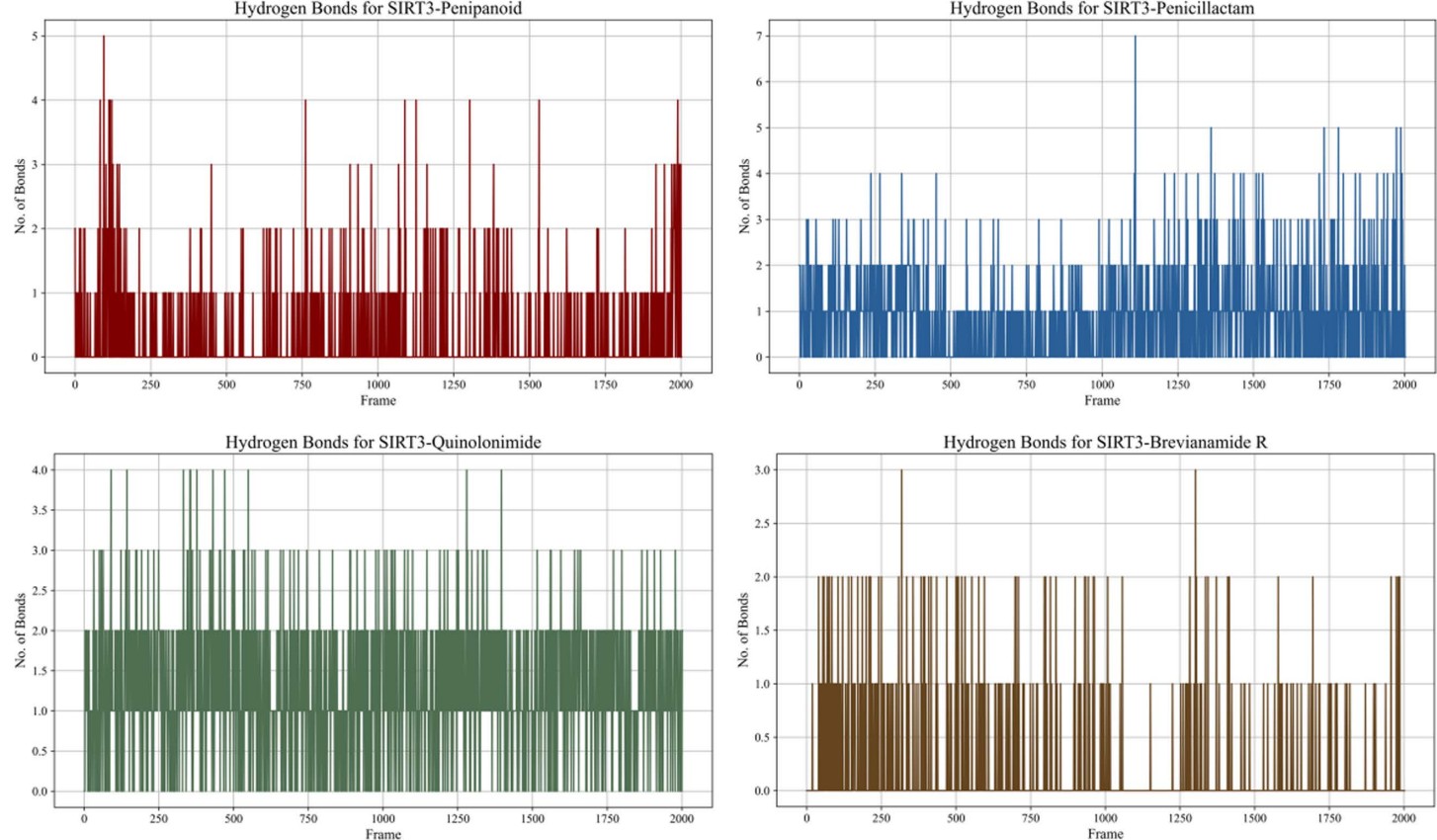

**Fig 7. Hydrogen bond formation over time for each SIRT3–ligands complex during the 100 ns MD simulation.** Plots (from top-left to bottom-right) show Penipanoid C, Penicillactam, Quinolonimide, and Brevianamide R, respectively. The y-axis indicates the number of hydrogen bonds formed at each simulation frame (x-axis), illustrating how consistently each compound maintains polar contacts with the SIRT3 active site.

Phe157, His248, and Phe294. These findings underscore the critical role of the aromatic/hydrophobic pocket formed by Phe157, Phe180, Ile230, His248, and Phe294 in ligand binding, corroborating the stable interactions observed in both docking and MD simulations (**Fig 14**) [40].

## 4. Discussion

SIRT3, the primary deacetylase located within mitochondria, is implicated in various facets of mitochondrial metabolism, encompassing mitochondrial synthesis and movement. Given the critical role of mitochondria in cellular functions, any dysregulation leading to dysfunction can contribute to a wide array of diseases. SIRT3 dysregulation has been associated with different mitochondrial disorders. The amount of evidence that has accumulated over time has gradually revealed the complexities of SIRT3's innate biological functions, providing insight into possible therapeutic uses in human diseases. From its original classification as a gene associated with longevity to its current status as a "superstar" target for the treatment of multiple diseases, SIRT3 has attracted a lot of attention [41,42]. The focus of this study is to explore deep-sea fungal metabolites with the aim of identifying novel inhibitors against SIRT3, contributing to the ongoing efforts to understand and potentially manipulate SIRT3 for therapeutic purposes.

Microbial natural product research has gained prominence owing to the capacity of microorganisms to generate biosynthetically rare, biologically active, and structurally diverse metabolites like their hosts. The advantageous features of large-scale metabolite production without harming hosts, rapid generation, and reproducibility make this area of study

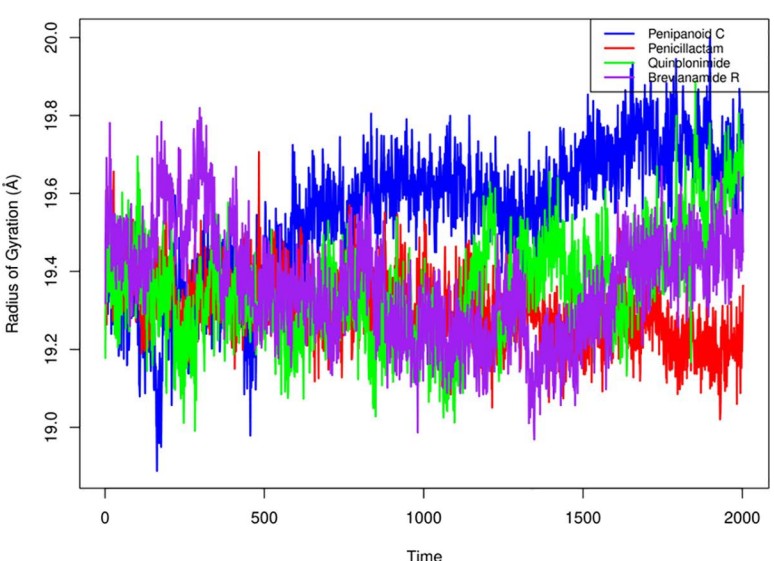

**Fig 8. Radius of gyration plot of all four complexes with respect to time.**

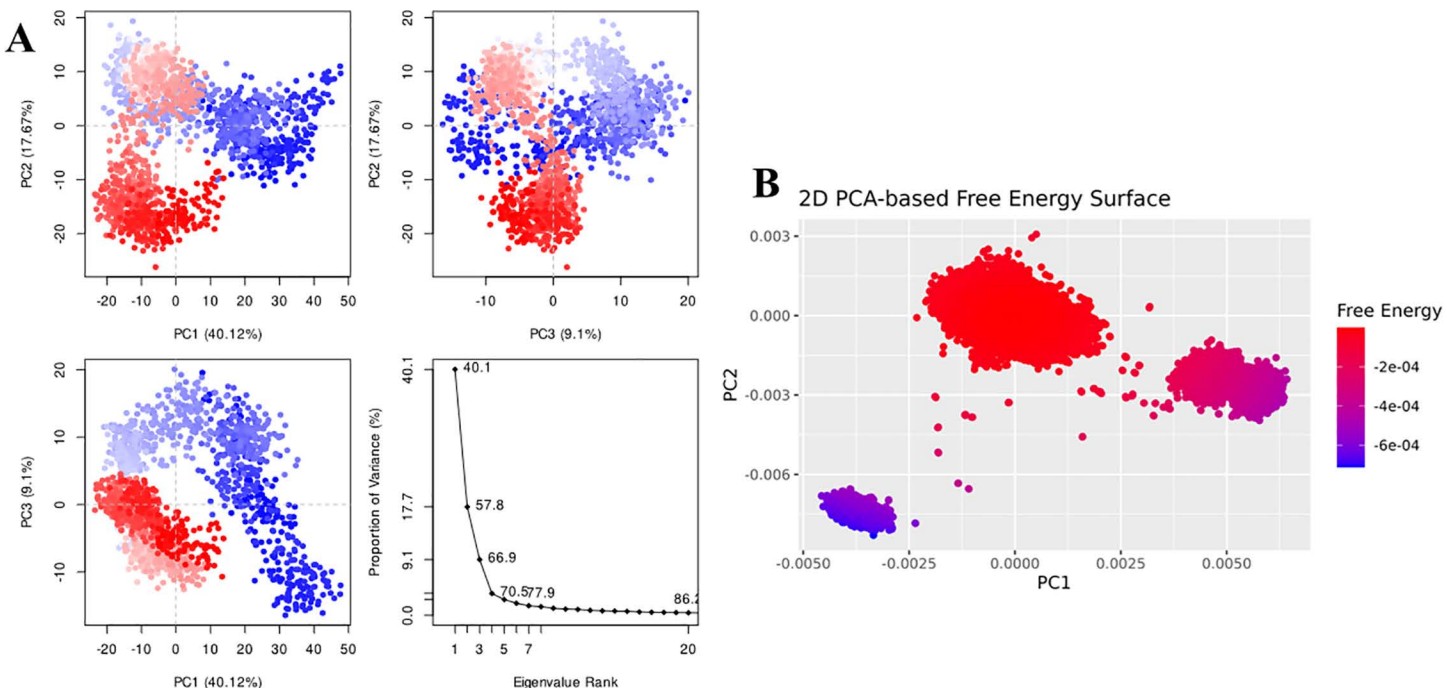

**Fig 9. (A) The principal component analysis for Penipanoid C complex, indicating the fluctuations in different hyperspaces.** (B) PCA based free energy surface of the complex calculated during simulation.

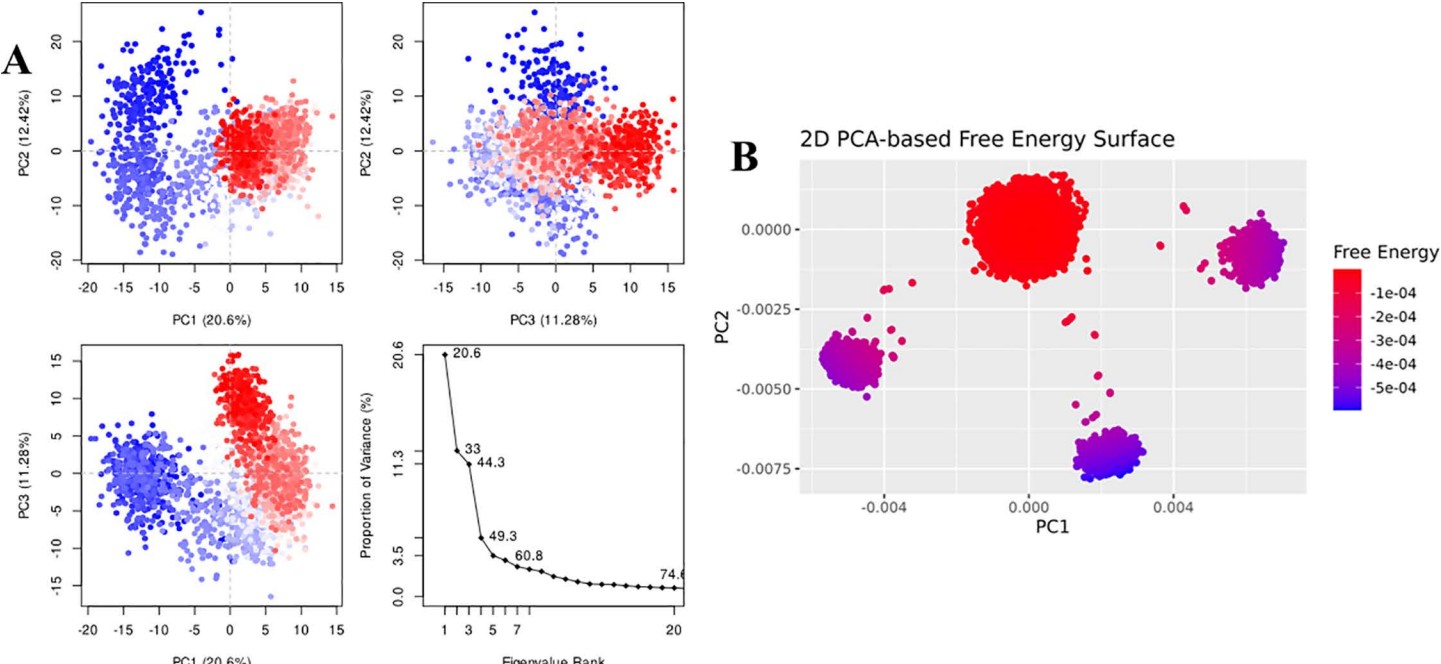

**Fig 10. (A) The principal component analysis for Penicilactam complex, indicating the fluctuations in different hyperspaces.** (B) PCA based free energy surface of the complex calculated during simulation.

particularly appealing. As a result, secondary metabolites derived from marine microorganisms exhibit distinctive and diverse structures, coupled with significant biological activities, holding substantial potential for applications in traditional Chinese medicine [43]. Deep-sea fungi have become a focal point for researchers due to their capacity to produce diverse secondary metabolites, high occurrence rates, and notable biological activities within the deep-sea microbial community. The identification of secondary metabolites from marine fungi is increasingly capturing the attention of researchers seeking to develop novel drugs [44,45].

The structures of deep-sea fungal metabolites were obtained from PubChem. The prepared compounds were docked to the prepared SIRT3 receptor to predict binding affinities using the standard precision mode of the glide tool. The computational technique of molecular docking predicts the binding affinity of ligands to receptor proteins. Although it has applications in nutraceutical research, it has evolved into a formidable drug development tool[46,47]. The top ten compounds (Cyclopiamide D, Penilumamide K, Coccoquinone A, Penipanoid C, Penicillactam, Quinolonimide, Prenylterphenyllin, Brevianamide R, Butanolide A, and Dichotocejpin C) were chosen for analysis based on their binding affinities. The binding affinities of the chosen compounds ranged from -9.693 to -8.327 kcal/mol. The binding affinities of the selected compounds suggested that they could inhibit the function of the SIRT3 protein.

The assessment of ADMET characteristics and toxicity risks for the selected compounds revealed predicted values within an acceptable range. Evaluating the ADMET properties is a crucial aspect of the drug development process. It enables the prediction of potential toxicity, behavior, and outcomes of a proposed drug within the human body. Understanding these properties aids in making informed decisions about the safety and efficacy of drug candidates during the development phase [48–50]. Except for Cyclopiamide D, Coccoquinone A, Butanolide A, and Dichotocejpin C, none of the compounds were found to be toxic. Based on the pharmacokinetic and toxicity profiles, four compounds were chosen for molecular interactions analysis Penipanoid C, Penicillactam, Quinolonimide, and Brevianamide R.

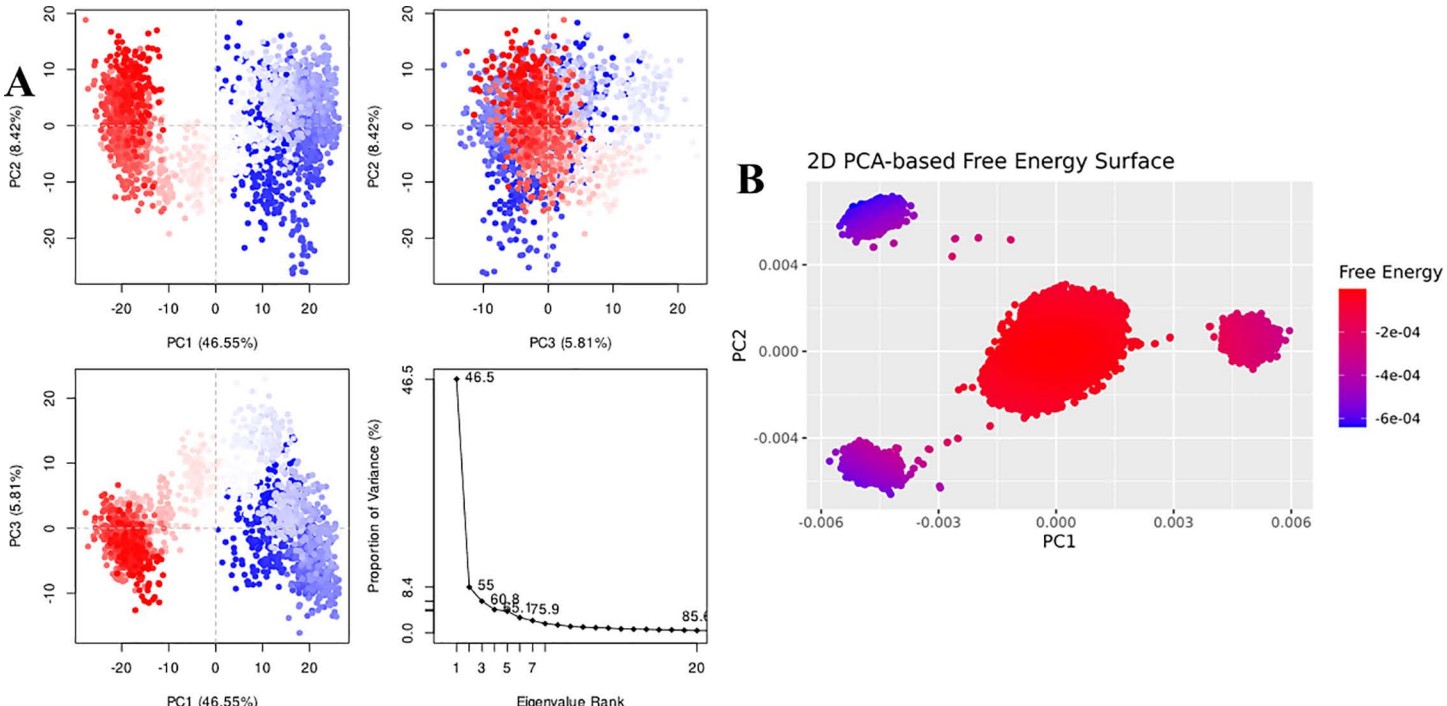

**Fig 11. (A) The principal component analysis for Quinolinimide complex, indicating the fluctuations in different hyperspaces.** (B) PCA based free energy surface of the complex calculated during simulation.

After the analysis of the molecular interactions, the compounds that were chosen for analysis were aligned on the co-crystal ligand to determine their binding modes. The plausible binding modes of the four hits were examined with alignment of the co-crystal ligand. Consequently, the aligned modes were subjected to MD simulation for the protein dynamics and structure stability analysis. MD simulations serve as an effective tool for understanding the stability of protein-ligand complexes [51]. The MD simulation analysis suggests that the selected compounds remained stably bound to the target protein.

Researchers can observe the binding process in real time using MD simulations. MD can aid in the identification of potential drug candidates by monitoring the conformational changes of the protein and ligand, the binding pathway, and binding affinities. By offering a more thorough comprehension of the interactions between drug candidates and their target biomolecules, they facilitate the logical design of innovative therapeutics [51,52]. According to MD simulation studies, these compounds were stable as effective inhibitors within the protein binding pocket. Moreover, the hydrogen bonding analysis, radius of gyration, PCA (with its 2D free energy surface), per-residue energy decomposition, and MMGBSA calculations collectively confirm that our selected compounds form stable, energetically favorable interactions within the SIRT3 binding site. These integrated findings strongly support the potential of these deep-sea fungal metabolites as robust lead candidates for further experimental validation and therapeutic development. Overall, our findings underscore the promising potential of deep-sea fungal metabolites as effective modulators of SIRT3, suggesting that these compounds could be further developed into innovative therapeutic agents. Given SIRT3's critical role in maintaining mitochondrial and metabolic homeostasis, these leads represent valuable starting points for new treatments targeting diseases associated with SIRT3 dysregulation.

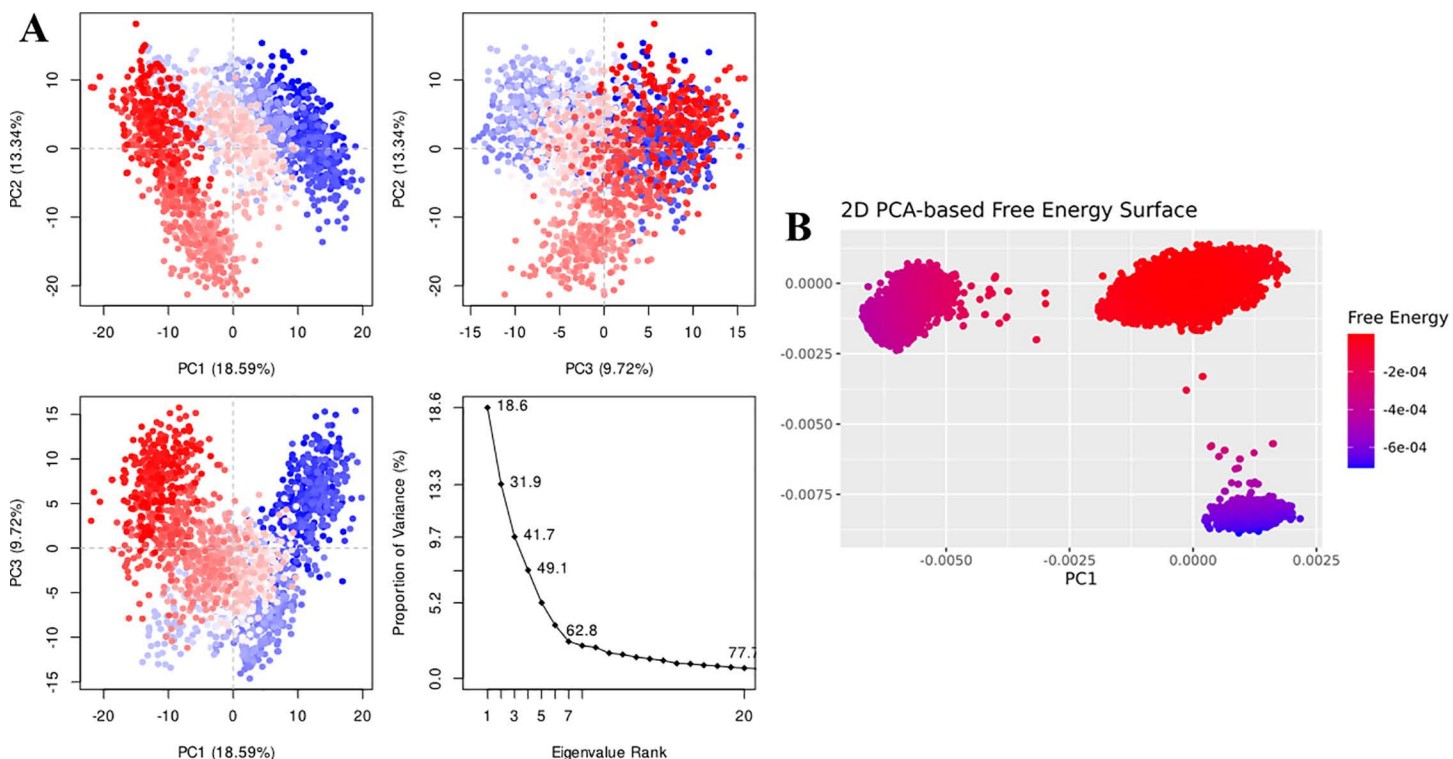

**Fig 12. (A) The principal component analysis for Brevianamide R complex, indicating the fluctuations in different hyperspaces.** (B) PCA based free energy surface of the complex calculated during simulation.

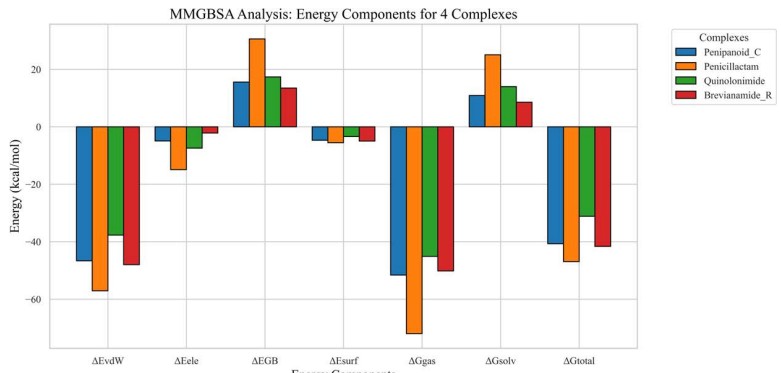

**Fig 13. MMGBSA energy component decomposition for the four SIRT3–ligand complexes.** Each bar represents a different energy contribution (van der Waals, electrostatic, polar and nonpolar solvation, and total binding energy) to the overall binding affinity. More negative values indicate stronger stabilization.

## Conclusion

In this study, we screened deep sea fungal metabolites for novel SIRT3 inhibitors using molecular docking and ADMET analysis to develop therapeutic interventions. The binding affinities of the screened hits were determined by molecular docking which indicated the strong interactions between ligand and protein. Further, the protein dynamics and

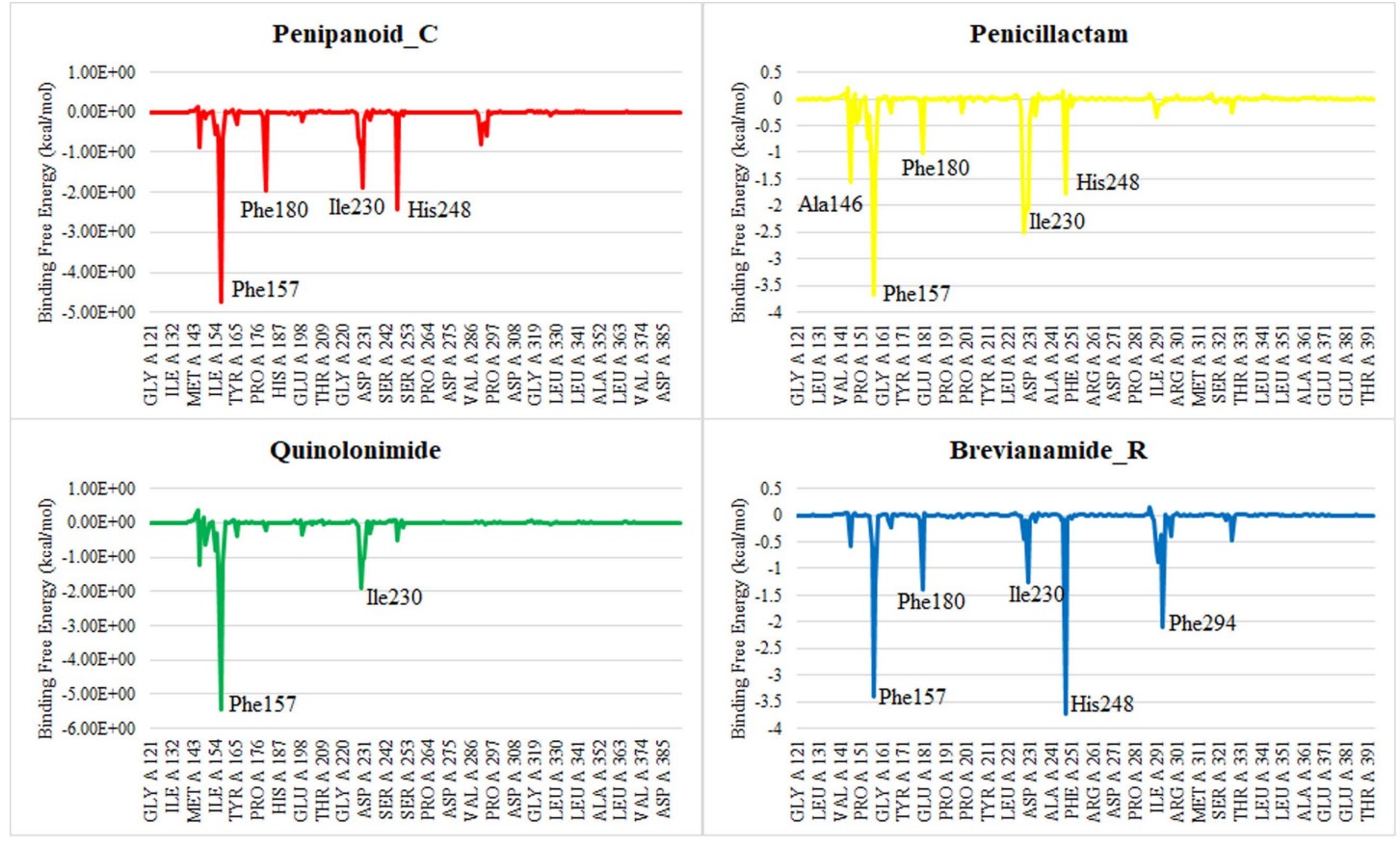

**Fig 14. Per residue energy decomposition of the four SIRT3-ligand complexes.** Bars below zero (negative values) indicate residues that strongly stabilize ligand binding, while positive values represent unfavorable contributions.

confirmational changes of the four selected compounds Penipanoid C, Penicillactam, Quinolonimide, and Brevianamide R, were analyzed by MD simulation which showed no major changes in the protein and stability of bound ligands with proteins. This promising outcome underscores the significance of further investigations and validations, both computationally and experimentally, to advance the understanding and development of these compounds as potential therapeutics.

## Author contributions

**Conceptualization:** Abdullah R. Alanzi.

**Data curation:** Bayan Abdullah Alhaidhal.

**Formal analysis:** Raghad Mohammad Aloatibi.

**Funding acquisition:** Abdullah R. Alanzi.

**Methodology:** Bayan Abdullah Alhaidhal.

**Supervision:** Abdullah R. Alanzi.

**Validation:** Raghad Mohammad Aloatibi.

**Writing – original draft:** Bayan Abdullah Alhaidhal, Raghad Mohammad Aloatibi.

**Writing – review & editing:** Abdullah R. Alanzi.

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
