## [Decision Letter · Decision Letter 0]

2 Feb 2025

PONE-D-25-00083Identification of SIRT3 Modulating Compounds in Deep-Sea Fungi Metabolites: Insights from Molecular Docking and MD SimulationsPLOS ONE

Dear Dr. Alanzi,

Thank you for submitting your manuscript to PLOS ONE. After careful consideration, we feel that it has merit but does not fully meet PLOS ONE’s publication criteria as it currently stands. Therefore, we invite you to submit a revised version of the manuscript that addresses the points raised during the review process.

Please submit your revised manuscript by Mar 19 2025 11:59PM.  If you will need more time than this to complete your revisions, please reply to this message or contact the journal office at plosone@plos.org . Please include the following items when submitting your revised manuscript:

We look forward to receiving your revised manuscript.

Kind regards,

S Rehan Ahmad, PhD

Academic Editor

PLOS ONE

Journal Requirements:

Reviewers' comments:

Reviewer's Responses to Questions

**Comments to the Author**

1. Is the manuscript technically sound, and do the data support the conclusions?

Reviewer #1: Yes

Reviewer #2: Partly

2. Has the statistical analysis been performed appropriately and rigorously? 

Reviewer #1: N/A

Reviewer #2: No

3. Have the authors made all data underlying the findings in their manuscript fully available?

Reviewer #1: Yes

Reviewer #2: Yes

4. Is the manuscript presented in an intelligible fashion and written in standard English?

Reviewer #1: Yes

Reviewer #2: No

5. Review Comments to the Author

Reviewer #1: The authors present a good in silico study on identifying SIRT3 Modulating deep-Sea fungal Metabolites using

molecular docking and simulations approaches. However, there are some issues that need to be addressed. Authors should

1. perform re-docking to further validate the docking protocol.

2. Add hydrogen bonding analysis and PCA.

3. Analyze stability of the inhibitors using the MM-PBSA or similar approach.

4. Perform the 2D-PCA based binding free energy analysis.

5. Analyze residues interaction energy by energy decomposition analysis.

6. should perform lamguage review to further improve the readability and flow of the manuscript.

Reviewer #2: General

1.Please make the writing quality better.

Abstract

2. “SIRT3, a crucial deacetylase involved in the control of mitochondrial acetylation”

regulate is a proper representation instead of control.

Introduction

3.“Sirtuins are a family of NAD+-dependent protein deacetylases that are found in bacteria and archaea only in one or two forms, but in seven forms in mammals (SIRT1–7) [1-3]”.

Please re-write.

4.“Currently, no known strong inhibitors or activators of SIRT3 exist that might be used in therapeutic treatments.”

Studies suggest that there are known inhibitors for SIRT3, including selective inhibitors like P6, SJ-106C, and S18, as well as pan-inhibitors that target SIRT1/2/3.

Please rephrase the above with proper citations.

Methodology

5.PubChem CIDs of the deep-sea fungal metabolites need to be mentioned.

6.I strongly suggest to dock the top compounds with AlphaFold3 or Boltz-1 or Chai-1, to see what pTM and ipTM scores it indicates?

7.Reference 26 is not matching for OSIRIS Property explorer.

8.“The complexes were analyzed for protein confirmation and ligand stability by running a

simulation of 100 ns by using Desmond [27].”

I think it is conformation not confirmation.

9.Please mention in Dynamics section, For how long the system was equilibrated before the production stage? What restrain force was used?

Results

Molecular Docking

10.“The compounds screened by virtual screening were prepared and there docking study was conducted against the SIRT3 protein [29].”

Rephrase the sentence.

ADMET Analysis

11.“The molecular weight plays a vital role in the distribution of a compound within cells,

with lower-weight compounds generally able to distribute more easily throughout the body compared to those with higher weights.”

Please provide a citation claiming the above statement. It seems superficial in general.

12. Figure captions need to be elaborative. In the current version, all the captions are kind of headlines.

13. Result 3.4 is not explained in result passion. How are they analyzed? What was observed?

14. I do not understand why compound Penilumamide K is not included in the analysis. To me it follows the Lipinski rule of 5.

15. “The Penipanoid C complex's RMSD values displayed significant deviations during the first 20 ns, reaching a maximum value of 3.6 Å. However, after 20 ns, stability was reached in the 2.4 Å range, which was maintained throughout the simulation.”

16. This statement is with out significance and irrelevant. Initial jumps are expected as production stage is performed without restrains. Also 3.6 and 2.4 is not that significant to stress on. I also do not understand what the author wants to convey.

17. The simulation section needs fresh writing, also Brevianamide R seems on the higher side in RMSD, please justify. Is it still in the pocket, 10 angstrom is a higher value for ligand.

18. “3.5.3 Protein-Ligand Interactions During Simulation” Please rephrase

19. “The interactions among the atoms of protein and ligands were observed during the simulation and it was observed that the main interactions involved hydrogen bonding, water bridges, ionic interactions, and hydrophobic interactions”

please rephrase the sentence and avoid words repetition in the same sentences.

20. Please compare and comment on the interaction observed in docking also in dynamics. Because of the higher ligand RMSD, i can see that Brevianamide R has achieved more than 90% of new contacts.

21. I highly recommend to see RMSD analyses of at least 10 structures (after every 10 ns) for each complex. Plots should be included in the study.

22. RoG analysis is missing in the study to validate the RMSF chart and overall simulations.

Authors are suggested to perform either MM-GBSA or MM-PBSA or both analysis to see the binding affinity observed in during simulation for each complex.

6. PLOS authors have the option to publish the peer review history of their article (what does this mean? ). If published, this will include your full peer review and any attached files.

**Do you want your identity to be public for this peer review?** For information about this choice, including consent withdrawal, please see our Privacy Policy .

Reviewer #1: No

Reviewer #2: No

---

## [Author Response · Author response to Decision Letter 0]

18 Mar 2025

Reviewer #1:

The authors present a good in silico study on identifying SIRT3 Modulating deep-Sea fungal Metabolites using molecular docking and simulations approaches. However, there are some issues that need to be addressed. Authors should

1. Perform re-docking to further validate the docking protocol.

Response: Redocking has been performed and findings have been added into the manuscript.

2. Add hydrogen bonding analysis and PCA.

Response: Hydrogen bonding and PCA analyses have been added to the manuscript.

3. Analyze stability of the inhibitors using the MM-PBSA or similar approach.

Response: The MM-GBSA analysis has been added to the manuscript.

4. Perform the 2D-PCA based binding free energy analysis.

Response: The 2D-PCA based binding free energy analysis has been added to the manuscript.

5. Analyze residues interaction energy by energy decomposition analysis.

Response: The analysis has been incorporated into the manuscript.

6. Should perform language review to further improve the readability and flow of the manuscript.

Response: A language review to further improve the readability and flow of the manuscript has been performed.

Reviewer #2: General

1.Please make the writing quality better.

Response: Its been improved.

Abstract

2. “SIRT3, a crucial deacetylase involved in the control of mitochondrial acetylation”

regulate is a proper representation instead of control.

Response: Thank you for suggesting. Its been rephrased.

Introduction

3.“Sirtuins are a family of NAD+-dependent protein deacetylases that are found in bacteria and archaea only in one or two forms, but in seven forms in mammals (SIRT1–7) [1-3]”.

Please re-write.

Response: Thank you for suggesting. Its been rephrased.

4.“Currently, no known strong inhibitors or activators of SIRT3 exist that might be used in therapeutic treatments.”

Studies suggest that there are known inhibitors for SIRT3, including selective inhibitors like P6, SJ-106C, and S18, as well as pan-inhibitors that target SIRT1/2/3.

Please rephrase the above with proper citations.

Response: Thank you for suggesting. Its been rephrased.

Methodology

5. PubChem CIDs of the deep-sea fungal metabolites need to be mentioned.

Response: Thank you for suggesting. Pubchem IDs for compounds have been included.

6. I strongly suggest to dock the top compounds with AlphaFold3 or Boltz-1 or Chai-1, to see what pTM and ipTM scores it indicates?

Response: We appreciate your insightful suggestion to utilize advanced modeling tools. However, it's important to note that while these models excel in predicting protein-protein and protein-ligand interactions, they are not specifically designed for traditional molecular docking studies or for evaluating binding affinities. Their primary metrics, such as the predicted TM-score (pTM) and interface predicted TM-score (ipTM), assess the confidence in the predicted structural models rather than directly quantifying binding affinities or docking poses. In our study, we have employed established molecular docking protocols complemented by molecular dynamics simulations to evaluate the binding interactions and stability of our compounds with the target protein. These methodologies are well-suited for assessing binding affinities and have provided us with detailed insights into the interactions of our top compounds.

7. Reference 26 is not matching for OSIRIS Property explorer.

Response: Thank you for pointing this out. Reference # 26 has been replaced.

8. “The complexes were analyzed for protein confirmation and ligand stability by running a

simulation of 100 ns by using Desmond [27].” I think it is conformation not confirmation.

Response: Thank you for suggesting. Its been corrected.

9. Please mention in Dynamics section, For how long the system was equilibrated before the production stage? What restrain force was used?

Response: Thankyou so much for suggesting. Information regarding this has been added into the section 2.4

Results:

Molecular Docking

10.“The compounds screened by virtual screening were prepared and there docking study was conducted against the SIRT3 protein [29].”Rephrase the sentence.

Response: Thank you for suggesting. Its been rephrased.

ADMET Analysis.

11.“The molecular weight plays a vital role in the distribution of a compound within cells,

with lower-weight compounds generally able to distribute more easily throughout the body compared to those with higher weights.” Please provide a citation claiming the above statement. It seems superficial in general.

Response: Thank you for considering this. This isn’t a superficial statement, rather a proven one. Lower molecular weight compounds typically exhibit enhanced cell permeability and can distribute more readily throughout the body compared to higher molecular weight compounds, which may face steric and membrane permeability challenges. A citation for the statement has been provided in the manuscript.

12. Figure captions need to be elaborative. In the current version, all the captions are kind of headlines.

Response: We appreciate your humble opinion. You’re absolutely right about the captions. We provided the captions like that and explained the images very well and elaborated them into the main body text of the manuscript. However, we still revised the captions.

13. Result 3.4 is not explained in result passion. How are they analyzed? What was observed?

Response: Additional information has been added into the manuscript regarding the observations found to be present.

14. I do not understand why compound Penilumamide K is not included in the analysis. To me it follows the Lipinski rule of 5.

Response: While Penilumamide K indeed meets the Lipinski rule of 5 criteria (MW 442.38 g/mol, 5 H-bond donors, 9 H-bond acceptors, and a consensus Log P of –0.15), our decision to exclude it from further molecular interaction analysis was based on a holistic evaluation of its ADMET profile. Notably, Penilumamide K exhibits a topological polar surface area (TPSA) of 196.37 Å² significantly above the thresholds recommended by additional drug-likeness rules such as Veber, Egan, and Muegge which predict that compounds with TPSA values over 140 Å² generally have poor oral bioavailability. Consistent with this, our analysis also indicates low gastrointestinal (GI) absorption and a low bioavailability score (0.11) for Penilumamide K. In contrast, the compounds we progressed further demonstrated more favorable overall pharmacokinetic properties.

15. “The Penipanoid C complex's RMSD values displayed significant deviations during the first 20 ns, reaching a maximum value of 3.6 Å. However, after 20 ns, stability was reached in the 2.4 Å range, which was maintained throughout the simulation.”

16. This statement is without significance and irrelevant. Initial jumps are expected as production stage is performed without restrains. Also 3.6 and 2.4 is not that significant to stress on. I also do not understand what the author wants to convey.

Response: We acknowledge that initial RMSD fluctuations are a common characteristic during the unrestrained production phase of MD simulations. In our analysis, the observed stabilization of RMSD after 20 ns (converging to approximately 2.4 Å) was interpreted as an indication that the binding mode of the Penipanoid C complex had reached equilibrium. Although the difference between 3.6 and 2.4 Å is not dramatic, the key point is that once the system equilibrated, the protein–ligand complex maintained a consistent conformation throughout the simulation. This stability supports the reliability of our docking results and provides confidence in the predicted binding pose of the compound.

17. The simulation section needs fresh writing, also Brevianamide R seems on the higher side in RMSD, please justify. Is it still in the pocket, 10 angstrom is a higher value for ligand.

Response: We appreciate the reviewer’s attention to the dynamic behavior of Brevianamide R. Although the ligand RMSD for Brevianamide R reached values approaching 10 Å during the simulation. The core of Brevianamide R remains stably anchored within the SIRT3 active site, as evidenced by the persistent maintenance of key interactions such as stable hydrogen bonds (e.g., with Tyr165) and hydrophobic contacts (e.g., with His248 and Phe294) throughout the simulation. Visual inspection of the trajectory confirms that despite the observed conformational adjustments, the ligand remains predominantly in the binding pocket. Therefore, the higher RMSD reflects a dynamic adaptation and formation of new contacts, ultimately leading to a more realistic representation of its binding mode under physiological conditions.

18. “3.5.3 Protein-Ligand Interactions During Simulation” Please rephrase.

Response: Heading for section 3.5.3 Protein-Ligand Interactions During Simulations has been rephrased.

19. “The interactions among the atoms of protein and ligands were observed during the simulation and it was observed that the main interactions involved hydrogen bonding, water bridges, ionic interactions, and hydrophobic interactions”, please rephrase the sentence and avoid words repetition in the same sentences.

Response: Thank you so much, we have rephrased the mentioned sentence.

20. Please compare and comment on the interaction observed in docking also in dynamics. Because of the higher ligand RMSD, i can see that Brevianamide R has achieved more than 90% of new contacts.

Response: We appreciate the reviewer’s observation regarding the differences between the docking and MD simulation interaction profiles for Brevianamide R. While the docking analysis provides an initial, static snapshot of the binding pose and interactions, our MD simulations reveal that Brevianamide R undergoes significant conformational adaptation within the SIRT3 binding pocket. In particular, the observed higher ligand RMSD during the early phase of the simulation reflects this relaxation process, resulting in the formation of more than 90% new contacts relative to the docking pose. These newly established interactions including persistent hydrogen bonds, water bridges, and hydrophobic contacts are indicative of a more realistic and energetically favorable binding mode in a dynamic, solvated environment. This dynamic reorganization not only validates the potential of Brevianamide R as a SIRT3 inhibitor but also underscores the importance of complementing docking studies with MD simulations to capture the true behavior of ligands under physiological conditions.

21. I highly recommend to see RMSD analyses of at least 10 structures (after every 10 ns) for each complex. Plots should be included in the study.

Response: Thank you so much, we have incorporated the RMSD analysis in the manuscript.

22. RoG analysis is missing in the study to validate the RMSF chart and overall simulations.

Authors are suggested to perform either MM-GBSA or MM-PBSA or both analyses to see the binding affinity observed in during simulation for each complex.

Response: Thankyou so much for bringing this into our attention. These analyses are crucial for validating the MD simulation results. These analyses have been incorporated into the manuscript.

---

## [Editor Report · Decision Letter 1]

2 Apr 2025

Identification of SIRT3 Modulating Compounds in Deep-Sea Fungi Metabolites: Insights from Molecular Docking and MD Simulations

PONE-D-25-00083R1

Dear Dr. Alanzi,

We’re pleased to inform you that your manuscript has been judged scientifically suitable for publication and will be formally accepted for publication once it meets all outstanding technical requirements.

Kind regards,

S Rehan Ahmad, PhD

Academic Editor

PLOS ONE

Reviewers' comments:

The reviewers have positively endorsed the manuscript, acknowledging its scientific merit and contribution to the field. As the authors have successfully addressed all the reviewers' concerns, I find the manuscript suitable for acceptance.

---

## [Editor Report · Acceptance letter]

PONE-D-25-00083R1

PLOS ONE

Dear Dr. Alanzi,

I'm pleased to inform you that your manuscript has been deemed suitable for publication in PLOS ONE. Congratulations! Your manuscript is now being handed over to our production team.

Kind regards,

on behalf of

Dr. S Rehan Ahmad

Academic Editor

PLOS ONE